# Long-term foot outcomes following differential abatement of inflammation and osteoclastogenesis for active Charcot neuroarthropathy in diabetes mellitus

Liza Das[1], Ashu Rastogi[1]*, Edward B. Jude[2]*, Mahesh Prakash[3], Pinaki Dutta[1], Anil Bhansali[1]

1 Department of Endocrinology, PGIMER, Chandigarh, India, 2 Tameside and Glossop Integrated Care NHS Foundation Trust, Ashton on Lyne, United Kingdom, 3 Department of Radiodiagnosis, PGIMER, Chandigarh, India

* Edward.jude@tgh.nhs.uk (EBJ); ashuendo@gmail.com (AR)

## Abstract

**Data Availability Statement:** All relevant data are within the manuscript and its Supporting Information files.

### Aims

Inflammatory osteolysis is sine-qua-non of active Charcot neuroarthropathy (CN) causing decreased foot bone mineral density (BMD) and fractures. We aimed to explore the effect of anti-inflammatory or anti-resorptive agents for effect on foot bone mineral content (BMC) and consequent long-term outcomes of foot deformities, fractures and amputation.

### Methods

Forty-three patients with active CN (temperature difference >2°C from normal foot) were evaluated. Patients were off-loaded with total contact cast and randomized to receive either methylprednisolone (1gm) (group A), zoledronate (5mg) (group B) or placebo (100ml normal saline) (group C) once monthly infusion for three consecutive months. Change in foot BMC was assessed at 6 months or at remission and followed subsequently up to 4 years for the incidence of new-onset fracture, deformities, or CN recurrence.

### Results

Thirty-six participants (24 male, 12 female) were randomized (11 in group A, 12 group B, 13 group C). The mean age was 57.7± 9.9 years, duration of diabetes 12.3± 5.8 years and symptom duration 6.5± 2.8 weeks. BMC increased by 36% with zoledronate (p = 0.02) but reduced by 13% with methylprednisolone (p = 0.03) and 9% (p = 0.09) with placebo at remission. There were no incident foot fractures, however, two patients sustained ulcers, and 3 had new-onset or worsening deformities and none required amputation during 3.36 ± 0.89 years of follow-up.

**Funding:** The author(s) received no specific funding for this work.

**Competing interests:** The authors have declared that no competing interests exist.

**Abbreviations:** AGE, Advanced glycation end-products; BMC, Bone mineral content; BMD, Bone mineral density; CGRP, Calcitonin gene related peptide; CN, Charcot neuroarthropathy; BTMs, Bone turnover markers; CTX, Carboxy terminal collagen crosslinks; DEXA, Dual Energy X-ray absorptiometry; DNS, Diabetes neuropathy score; ECLIA, Electrochemiluminescence; eGFR, estimated Glomerular filtration rate; ELISA, Enzyme linked immunosorbent assay; ESR, Erythrocyte sedimentation rate; hsCRP, high sensitivity C reactive protein; IL-1β, Interleukin-1 beta; IL-6, Interleukin 6; OPG, Osteoprotegerin; NDS, Neuropathy disability score; P1NP, Procollagen I intact N-terminal propeptide; RA, Rheumatoid arthritis; RANKL, Receptor activator of NF-κB ligand; TNF-α, Tumor necrosis factor-alpha; TCC, Total contact cast; VPT, Vibration perception threshold.

## Conclusion

Bisphosphonate for active CN is associated with an increase in foot bone mineral content as compared to decrease with steroids or total contact cast but long-term outcomes of foot deformities, ulceration and amputation are similar.

## Trial registration

**ClinicalTrials.gov:** NCT03289338.

## Introduction

The etiopathogenesis of Charcot neuroarthropathy (CN) is intriguing since its early description in 1868 [1]. Early elucidation for the causation of CN with the neurotraumatic and neuro-vascular theories were accurate in their times [2, 3]. Of late, the conceptual understanding of CN has evolved after description of the role of osteoclastic resorption of foot bones by activation of receptor activator of nuclear factor kappa-B (RANK) [3, 4]. The activation of RANK by RANK ligand (RANKL) occurs as a result of recurrent trauma to an insensate foot inciting a pro-inflammatory cascade of multiple inflammatory cytokines locally, the most common being tumour necrosis factor-α (TNF-α), interleukin-1ß (IL-1β) and interleukin-6 resulting in a local 'cytokine storm' [5, 6]. In addition, non-inflammatory factors including hyperglycemic milieu by increasing advanced glycosylation end products (AGEs), and autonomic neuropathy by causing a decrease in calcitonin gene-related peptide (CGRP) and endothelial nitric oxide synthase, can upregulate the RANKL/ NF-κB pathway [7–9]. There is evidence to show genetic polymorphisms in RANKL-RANK-OPG genes and autoantibodies against post-translationally modified collagen to have a causal role for the development of CN [10, 11]. Thus, both the inflammatory and non-inflammatory pathways of RANK activation have been proposed for stimulating osteoclastogenesis in active CN of foot. At the same time, RANKL-independent osteoclastogenesis mediated through TNF-α and hyperglycemic milieu de novo has also been demonstrated in patients with CN [4, 12].

The gold standard treatment for the management of active CN is total contact cast (TCC) [13]. But TCC has inherent limitations including worsening of bone mineral density (BMD) [14, 15], cast-related tissue injury and prolonged immobilization usually for 6 to 12 months) [16]. Anti-resorptive agents that target osteoclastogenesis by RANK inhibition are logical therapeutic agents for the cessation of resorption in active CN. Previously, bisphosphonates like pamidronate [17], alendronate [18], zoledronate (ZL) [19, 20], calcitonin [21] and denosumab [22] have been evaluated for their role in active CN. Studies with these agents have evoked mixed results, as few earlier studies demonstrated benefits with calcitonin and pamidronate mostly in terms of parameters like symptom score and bone turnover markers (BTMs); but importantly, time to remission or total immobilization time were not evaluated [17]. Alendronate showed a non-significant effect on temperature difference [18] while Zoledronate showed significantly prolonged time to clinical remission as compared to placebo [19].

Therefore, considering the recent understanding of etiopathogenesis of CN and significant limitations of prior studies, we conducted an interventional, three parallel-arm, the study with an anti-resorptive agent, anti-inflammatory agent and a placebo in addition to TCC as the primary treatment modality in patients with active CN [23]. Methylprednisolone was chosen as its use in rheumatoid arthritis (RA), which is also characterized by cytokine-induced

inflammatory osteolysis akin to CN has beneficial effect on clinical remission in RA despite its known adverse effects on bone metabolism [24, 25]. The role of anti-inflammatory agents for curtailment of inflammation in active CN, though proposed, was not evaluated prior to our study [4–6, 23, 26]. Zoledronate was included in the study protocol as data with ZL in active CN is still contentious despite it being the most potent bisphosphonate available (100 times more than pamidronate) [27]. We observed no difference in remission rates for active CN with zoledronate compared to placebo and increased duration to remission with steroids [23]. A recent systematic review also confirmed no added benefit of pharmacological agents for short term outcome defined as remission of active CN [28]. However, long term consequences of CN are pertinent clinical outcomes including the occurrence of deformities, fractures, amputations and mortality [29, 30].

We hereby report the differential effects of these agents on foot bone mineral content and long-term outcome of new-onset fractures, deformities and amputation. The insights into the pathophysiology of active CN following the use of anti-inflammatory or anti-resorptive agent is detailed.

## Patients and methods

The study enrolled participants attending the multi-disciplinary foot clinic at a tertiary care centre in India. A total of 143 participants with CN were diagnosed during the study period, 42 of whom qualified for the initial screening and 36 were finally recruited into the study protocol (Fig 1). Active CN of foot was defined clinically as a localised swelling, erythema and temperature difference exceeding 2˚C compared to a similar site on the opposite foot. This definition has been suggested and used in prior RCTs involving pharmacotherapeutic management of active CN [17, 19]. Chronic CN was considered, in the presence of fracture/ dislocation, peripheral neuropathy and/or prior history suggestive of active CN. Participants fulfilling the criteria for active CN in the setting of chronic CN were considered as "active on chronic" CN. Those with self-reported diabetes or those already on treatment were included. Exclusion criteria included presence of pedal ulcer, osteoporosis (T score <-2.5 at lumbar spine or hip), gout, active peptic ulcer disease, steroid intake in the last three months,

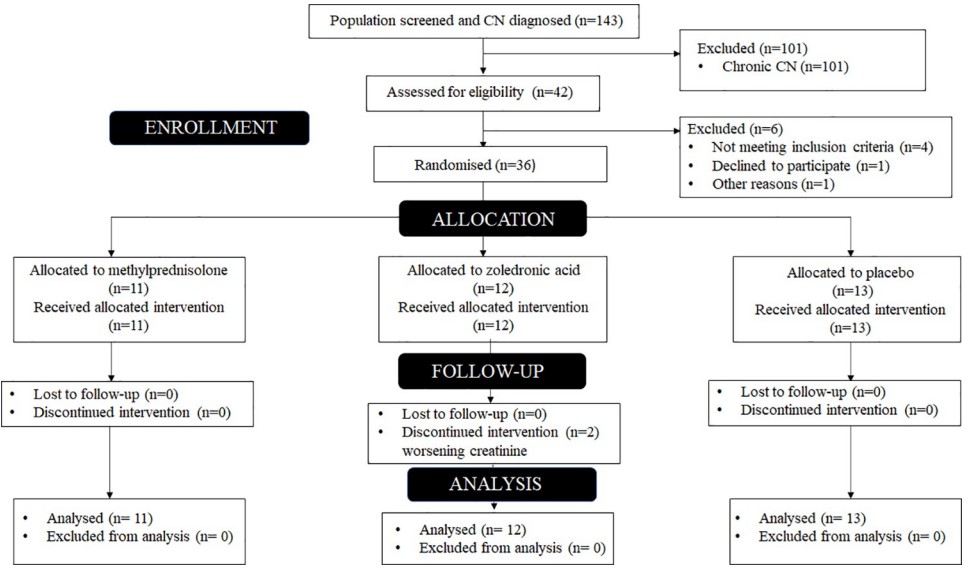

**Fig 1. Randomisation protocol as per the CONSORT guidelines (CN- Charcot's Neuroarthropathy).**

estimated glomerular filtration rate (eGFR) $< 45$ ml/min/m$^2$, active dental caries or invasive dental procedure, peripheral vascular disease (ABI $< 0.9$), bilateral foot involvement, pregnant/lactating women and those who had recently received antiresorptive agents (in the previous 12 months).

The diagnosis of active CN was further corroborated by X-ray and/or magnetic resonance imaging (MRI) (3T scanner Siemens MagnetromVerio). Sanders-Frykberg classification was used for anatomical grading and localization of the involved site of the foot. The study was approved by the Institutional Ethics committee (IEC/2016/2276) and written informed consent was obtained from all participants.

Clinical details regarding duration of symptoms, inciting event, diabetes duration and co-existing microvascular and macrovascular diabetic complications were recorded. Detailed neurological examination was performed including vibration perception threshold (VPT>25 mV was considered as abnormal) by biothesiometer—Vibrometer-VPT® (Diabetik Foot Care, Madras Engineering Service, India), 10-g monofilament (Diabetik Foot Care, Madras Engineering Service, India) perception at 5 standardized plantar sites and ankle reflex. Foot temperature was measured by infrared dermal thermometry (FLIR Systems Inc, Orlando, USA) with a pixel resolution of 4800 (80*60), thermal sensitivity of $<0.15$°C and range of detection from -20°C to 250°C. Bone mineral content (BMC) in gram and BMD (g/cm$^2$) at the region of interest (ROI) of foot (site of maximum temperature difference compared to opposite foot) were quantified with Dual Energy X-ray Absorptiometry (DEXA, Hologic 6.0, Model- Discovery A, S/N 87292). The involved foot was scanned from toes to heel [31]. ROI was drawn manually, and BMC quantified. Blood sample for biochemistry, bone turnover markers (BTMs) and inflammatory cytokines were collected after an overnight fast (8–10 hours) and analysed by ECLIA as detailed earlier [23].

All participants were provided with standardised non-walking, non-removable fibre-glass TCC for immobilisation. Subsequently they were randomised by one of the investigators (AR) using computer generated randomisation blocks of 3, to receive an infusion of methylprednisolone 1gm in 100ml normal saline (NS) (Group A), zoledronate 5mg in 100ml NS (Group B) or 100ml NS (placebo) (Group C) administered between 0900 to 1000h once a month for three consecutive months. All participants were followed fortnightly and change of cast was offered in view of 'pistoning' effect due to reduction of edema. An average of 3 temperature recordings at the ROI of foot was obtained after the removal of cast for 30 minutes, during each follow up visit. Inflammatory cytokines, BTMs and BMC were evaluated at baseline and at clinical remission or 6 months (whichever occurred earlier). Clinical remission of active CN was defined as a temperature difference $<2$°C between the affected foot and a similar site on the opposite foot on two successive follow-up visits two weeks apart [13, 32].

After clinical remission of active CN, the TCC was discontinued, and participants were provided with cast walker/ modified footwear for Charcot foot during subsequent follow up. The participants were reviewed three monthly with through foot examination for the recurrence of CN (foot temperature assessment), incidence of deformities, ulcers (clinical examination), new onset fractures (radiological assessment by blinded investigator), or amputation.

## Statistical analysis

Normality of the data for each variable was assessed by Kolmogorov-Smirnov test. Data are expressed as mean ± SD if normally distributed and as median and inter-quartile range if skewed. Student T-test was used to compare the means of two groups for parametric data and Mann-Whitney U test for non-parametric data. ANOVA was used for comparing means of the three groups for parametric data and Kruskal-Wallis test was used for non-parametric

data. A Kaplan-Meier curve was constructed to assess the difference in remission of active CN with the interventions. Cox proportion hazard model was used to identify the association between the baseline variables and incident remission. The variables were decided by their clinical relevance to active CN of foot. Pearson correlation analysis was performed to identify relation between change in BMC with the inflammatory cytokines and bone turnover markers. SPSS version 22 was used for data analysis and a p-value <0.05 was considered significant.

## Results

Forty-two participants were enrolled in the study. Six were excluded as detailed in Fig 1. The demographic details, diabetic complications and foot characteristics of the participants at study enrollment are shown in Table 1. The mean time for clinical remission in the whole cohort was 15.5 ± 4.2 weeks and was significantly higher in the methylprednisolone group as compared to either zoledronate or placebo groups (p = 0.01) (Fig 2) [23]. None of the baseline parameters was found to be associated with incident remission of active CN of foot (Table 2).

There was a 13% (p = 0.03) and 9% (0.09) reduction in BMC (ROI) with methylprednisolone and placebo, respectively, but 35.8% (p = 0.02) increase in the zoledronate group (Fig 3). There was no corelation between the change in BMC after intervention with the baseline inflammatory cytokines, BTMs or the changes observed in these parameters after intervention

**Table 1. Baseline clinical and biochemical parameters of the cohort.**

| Parameter | Group A | Group B | Group C | p value |
|---|---|---|---|---|
| | n = 11 | n = 12 | n = 13 | |
| Age (years) | 51.1 ± 4.7 | 60.9 ± 8.2 | 59.1 ± 12.4 | 0.05 |
| BMI (kg/m$^2$) | 26.1 ± 4.1 | 25.9 ± 5.2 | 24.9 ± 4.2 | 0.62 |
| Males: Females | 6: 5 | 8: 4 | 10: 3 | 0.38 |
| Duration of diabetes mellitus (years) | 10.9 ± 6.2 | 13.2 ± 2.6 | 12.1 ± 6.4 | 0.37 |
| Duration of symptoms (months) | 3.5 ± 2.1 | 2.1 ± 1.4 | 2.7 ± 1.8 | 0.19 |
| Insensate to monofilament (%) | 80 | 86 | 100 | 0.50 |
| Precipitating event (%) | 30 | 38.5 | 54 | 0.49 |
| Active disease (%) | 65 | 80 | 72 | 0.62 |
| Active on chronic disease (%) | 35 | 20 | 28 | |
| VPT (mv) | 32 | 29 | 30 | 0.46 |
| Right: left foot (%) | 60:40 | 57:43 | 67:33 | 0.93 |
| Midfoot involvement (%) (SF III) | 73 | 50 | 69 | 0.81 |
| Retinopathy (%) | 64 | 75 | 70 | 0.82 |
| Nephropathy (%) | 55 | 67 | 70 | 0.59 |
| HbA1c (mmol/mol) | 77 | 79.2 | 67.2 | 0.12 |
| HbA1c (%) | 9.2 ± 1.9 | 9.4 ± 1.4 | 8.3 ± 1.2 | 0.12 |
| eGFR (ml/min/m$^2$) | 73.3 ± 10.2 | 68.3 ± 17.7 | 68.2 ± 13.9 | 0.78 |
| Calcium (mg/dl) | 9.42±1.03 | 8.99±0.71 | 9.03±0.57 | 0.66 |
| 25(OH)D (ng/ml) | 20.08±1.06 | 25.36±0.98 | 19.38±1.43 | 0.90 |
| iPTH (pg/ml) | 40.06±2.98 | 35.21±5.13 | 48.05±2.33 | 0.76 |
| ESR (mm/hr) | 12 (11–15.75) | 13 (12–21) | 18 (14–21) | 0.10 |
| hsCRP (mg/L) | 3.7 (1.87–12.05) | 3.6 (2.5–28) | 4.0 (2.87–14.60) | 0.58 |
| Procalcitonin (ng/ml) | 0.04 (0.02–0.10) | 0.04 (0.02–0.10) | 0.02 (0.02–0.05) | 0.47 |

Group A- Methylprednisolone; Group B- Zoledronate; Group C- Placebo; SF- Sanders Frykberg classification of involvement of region/joint in Charcot foot. Data are expressed in Mean ± SD, percentage expressed in terms of the whole group or median (q25-q75) depending on normality.

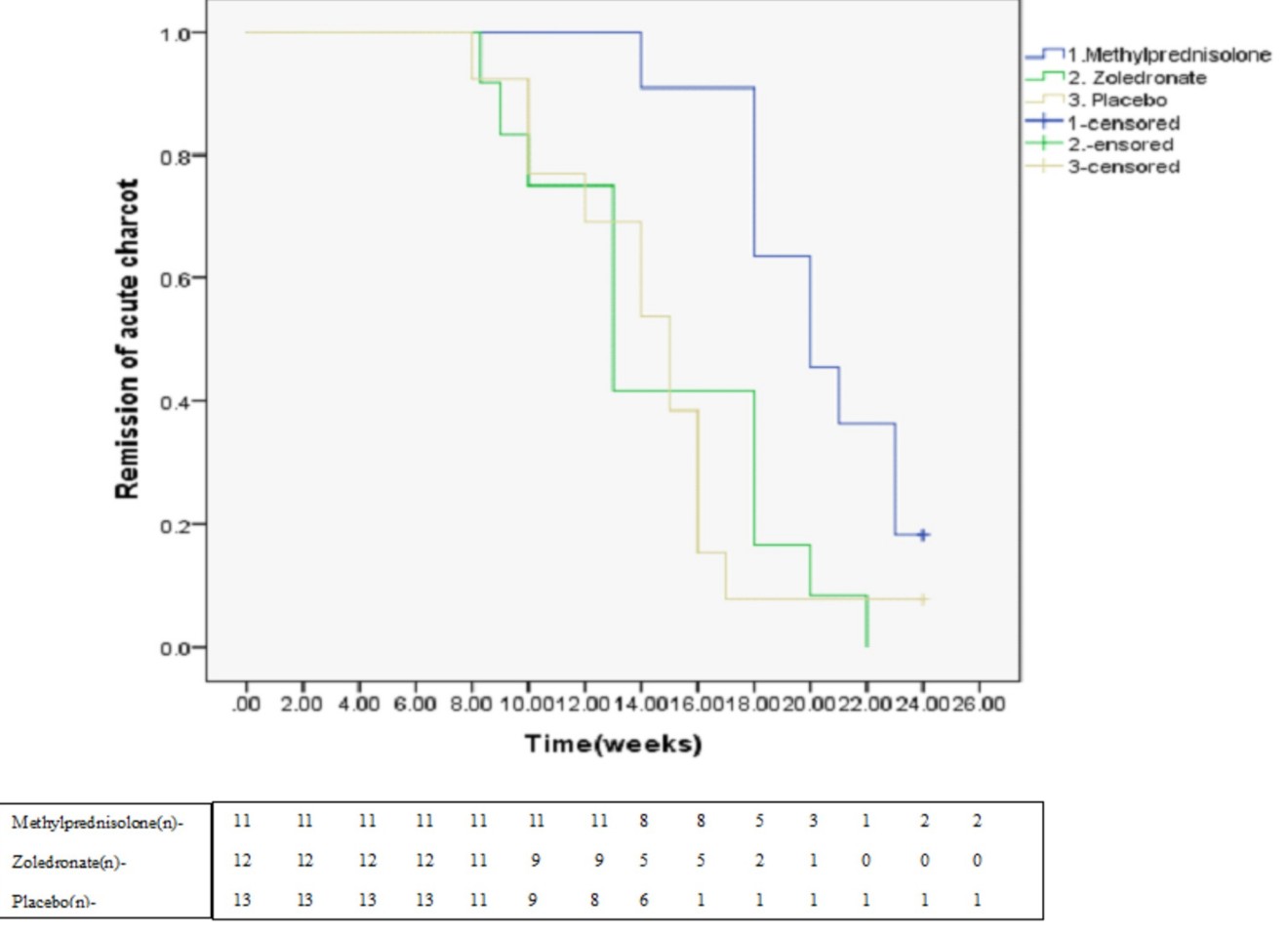

| Methylprednisolone(n)- | 11 | 11 | 11 | 11 | 11 | 11 | 11 | 8 | 8 | 5 | 3 | 1 | 2 | 2 |
| Zoledronate(n)- | 12 | 12 | 12 | 12 | 11 | 9 | 9 | 5 | 5 | 2 | 1 | 0 | 0 | 0 |
| Placebo(n)- | 13 | 13 | 13 | 13 | 11 | 9 | 8 | 6 | 1 | 1 | 1 | 1 | 1 | 1 |

**Fig 2. Kaplan-Meier curve for remission of active Charcot foot in the three groups.**

at 6 months (Table 3). Despite significant reduction in pro-inflammatory cytokines with MP, ongoing osteolysis could not be abated indicating the role of cytokine-independent pathways in the progression of bone destruction, as elaborated in Fig 4.

Dashed arrows indicate inhibition and solid arrow indicates activation of the particular target

○ Total contact cast- Standard of care for remission of active CN

○ Methylprednisolone- No proven benefit in causing remission of active CN

○ Bisphosphonates- Equivocal evidence in causing remission of active CN

○ Denosumab- Proven efficacy for early remission of active CN

○ Teriparatide- No available evidence in active CN but benefit shown in chronic CN

Adverse events included flu-like reaction (n = 5, 41.6%) and acute kidney injury (defined as increase in serum creatinine >0.5mg/dl above baseline or estimated GFR under 30ml/min/m$^2$) (n = 2, 16.6%) noted with the use of zoledronate [33]. Worsening of glycemic profile was observed with methylprednisolone. In addition, cast related tissue injury were observed in 2 patients each in all the three groups.

**Table 2. Predictors of remission of active Charcot foot according to the baseline variables.**

| Parameters | Hazard Ratio | 95% CI | p-value |
|---|---|---|---|
| Age (years) | 0.958 | 0.910–1.008 | 0.099 |
| Duration of Diabetes (years) | 1.007 | 0.886–1.145 | 0.911 |
| Gender | 0.821 | 0.271–2.483 | 0.726 |
| Body Mass Index (kg/m$^2$) | 1.019 | 0.906–1.147 | 0.750 |
| Duration of symptoms(months) | 1.343 | 1.018–1.773 | 0.37 |
| HbA1c(mmol/mol) | 0.981 | 0.786–1.225 | 0.868 |
| eGFR (ml/min/1.73m$^2$) | 1.014 | 0.975–1.054 | 0.492 |
| ESR (mm) | 0.987 | 0.942–1.033 | 0.574 |
| hsCRP (mg/L) | 1.003 | 0.953–1.056 | 0.911 |
| Procalcitonin (ng/mL) | 127.942 | 0.00–39775145.75 | 0.452 |
| Baseline Bone mineral content (ROI) | 1.045 | 0.970–1.127 | 0.247 |
| Baseline foot temperature difference at ROI (˚C) | 0.928 | 0.648–1.329 | 0.683 |

eGFR: estimated glomerular filtration rate; ESR: Erythrocyte sedimentation rate; hsCRP: high sensitive C-Reactive Protein; ROI: Region of interest.

Hazard Ratio (95%CI) obtained by Cox Regression model.

## Long-term outcomes

Patients were followed for a mean duration of 3.36 ± 0.89 years. The mean HbA1c at follow-up was 9.81 ± 2.36%. Three participants had new-onset or worsening deformities as two cases had blunting of medial longitudinal arch leading to midfoot collapse (one each in group A and B) and one case of 1st meta-tarso-phalangeal joint subluxation (group B)]. There were no incident foot fractures noted on follow-up in any group. However, two patients sustained neuropathic ulcers after 13- and 18-months following remission of active CN that healed over 8 weeks, and none required amputation. There was one case of recurrence of active CN (ipsilateral foot,

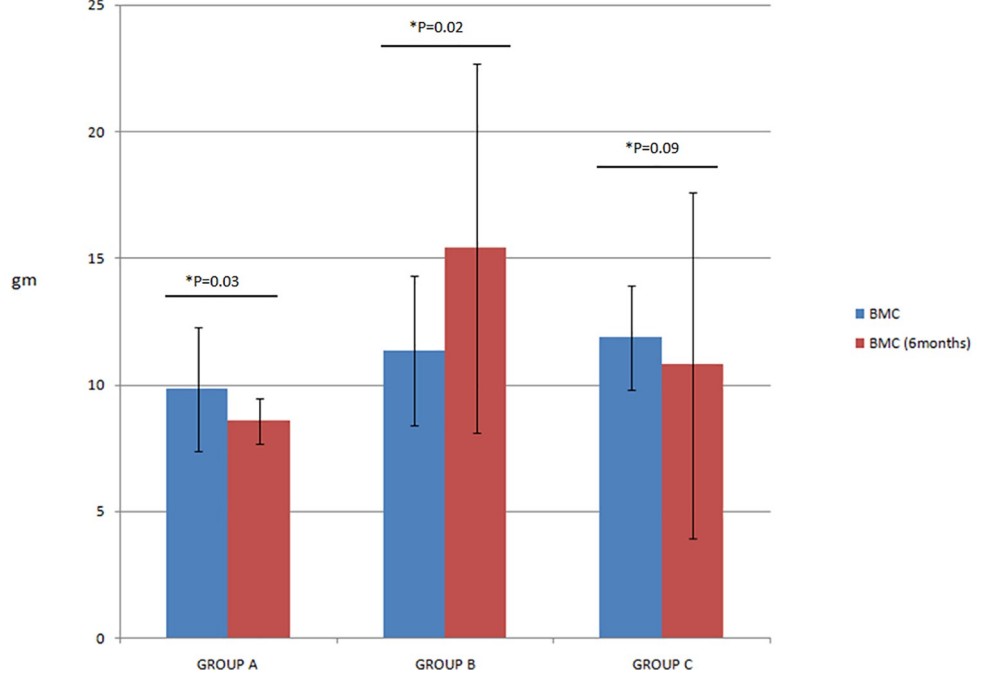

**Fig 3. Bone mineral content in study groups at randomization and after 6 months.** Group A- Methylprednisolone; Group B- Zoledronate; Group C- Placebo.

**Table 3. Correlation analysis between change in the foot mineral content with baseline inflammatory cytokines, bone turnover marker and their change at six month following intervention.**

| Parameters | R | p-value |
|---|---|---|
| Baseline PINP | -0.059 | 0.741 |
| Baseline CTX | -0.079 | 0.656 |
| Baseline TNF-α | 0.114 | 0.519 |
| Baseline IL-1β | -0.089 | 0.617 |
| Baseline temperature difference between feet | 0.075 | 0.675 |
| ΔP1NP | -0.014 | 0.938 |
| ΔCTX | -0.316 | 0.073 |
| ΔTNFα | -0.144 | 0.431 |
| Δ IL-1β | -0.040 | 0.830 |

r value obtained by Pearson correlation analysis.

Δ: change from baseline following intervention.

group B) without any preceding trauma after 46 months of initial intervention. Five patients died during follow-up (two in group B and 3 in group C) due to acute decompensated heart failure (one participant), coronary artery disease (two), chronic kidney disease (one) and one case of septicemia.

## Discussion

We observed that neither the use of MP nor ZL could translate into an early clinical remission as compared to TCC alone in active CN. There was a significant increase in foot BMC with

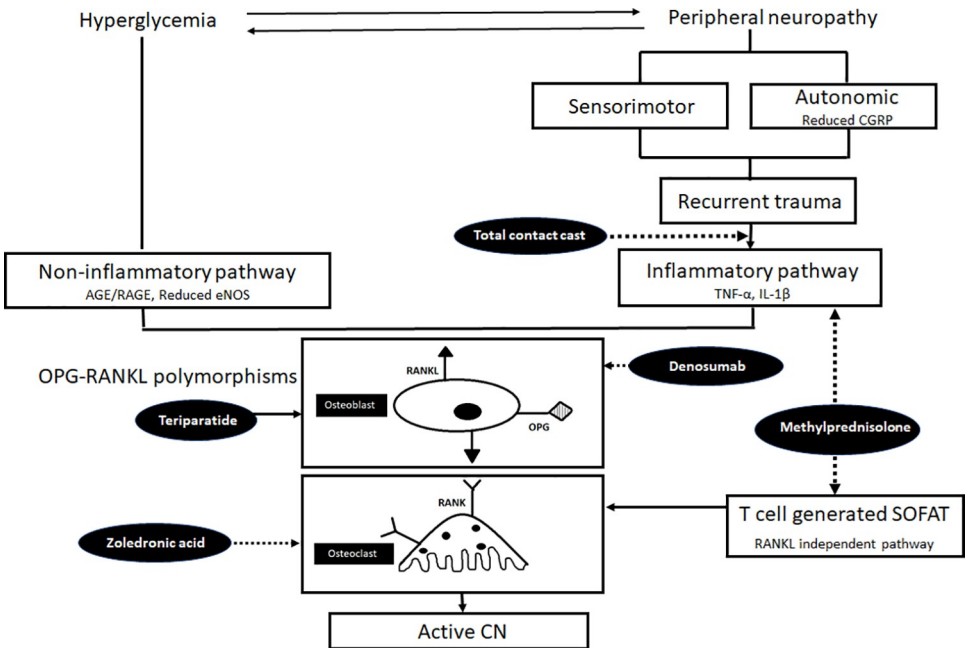

**Fig 4. Integrated pathophysiology and interactions between various factors in persons with diabetes implicated for the causation of active Charcot neuroarthropathy and efficacy of various evaluated therapeutic agents.** AGE-Advanced glycation end products; RAGE- Receptor for advanced glycation end products; eNOS- Endothelial nitric oxide synthase; SOFAT- secreted osteoclastogenic factor of activated T cells; RANKL- Receptor activator of nuclear factor-κβ, OPG- Osteoprotegerin; CGRP- Calcitonin gene related peptide; Charcot's neuroarthropathy (CN).

zoledronate compared to a decrease with methylprednisolone and placebo. However, there were no differences in incidence or progression of foot deformities, incident fractures or amputation during a follow up of nearly 4 years.

The criteria used to define remission of active CN are based on reduction of signs of inflammation including redness, swelling and normalisation of temperature difference ($<2°C$) between both feet, which is the most consistent and objective parameter. These clinical criteria usually correspond to resolution of marrow edema on T2W MRI images and normalisation of radiotracer uptake on TcMDP bone scan at ROI [34, 35]. The clinical signs of inflammation are predominantly contributed by preceding local 'cytokine storm' and consequent release of prostaglandins coupled with exaggerated vasoreactivity in response to cumulative minor trauma to an insensate foot. An ongoing osteolysis because of cytokine-mediated activation of osteoclastogenesis also contributes to signs of inflammation. Therefore, off-loading the inflamed foot in active CN with a non-walking TCC is considered as the "gold standard" of treatment [13, 36]. A non-walking TCC not only helps by preventing further trauma to the insensate foot thus abating inflammation and subsequent osteoclast activation but is also instrumental in reducing and favourably redistributing the abnormal plantar pressure distribution [37]. However, it usually takes more than 6–12 months for clinical remission [16, 38]. During this long period of non-ambulation, BMD of foot bones may be adversely affected [14, 15]. Moreover, the diminution of clinical signs of inflammation and cessation of ongoing osteolysis despite off-loading may not always be concordant and may persist for 6 to 12 months. Therefore, efforts are ongoing to identify pharmacotherapeutic agents that could target the pathophysiology of CN of foot in patients with diabetes.

The recent understanding of pathophysiology of active CN focuses on the role of inflammatory cytokines that are postulated to incite RANKL-NF-κB activation and consequently, local osteoclastogenesis in the affected foot bones [5, 6, 26]. There is some pre-clinical evidence demonstrating reduction in resorption with anti-TNF-α antibodies [39] but no clinical evidence for the use of anti-inflammatory agents in active CN, till date. The current study used methylprednisolone in high dose pulse therapy to reduce the local 'cytokine storm'. We observed a delay in resolution of active CN with MP, despite cogent suppression of cytokines. Although circulating cytokine levels were assessed, they were regarded as indicative of local cytokine concentration, owing to the fact they were assessed in the same individuals at baseline and end of follow-up. The delay in resolution of clinical activity of CN despite suppression of cytokines may be due to the uninhibited activation of cytokine-independent RANKL pathway; but also suppression of favourable anti-inflammatory cytokines involved in bone healing (IL-4, IL-10) and worsening of hyperglycemia which, in turn, directly exacerbates the RANKL-NF-κB activity and subsequent osteoclastogenesis [40, 41]. The increased and sustained bone resorption as the direct effect of steroids on osteoclasts and a decrease in bone formation resulting in ongoing osteolysis of foot bones could have contributed to a delay in clinical remission with steroids despite significant suppression of inflammation [42].

Lack of substantial evidence regarding efficacy of ZL in achieving clinical remission in active CN may be attributed to the doses used in previous studies (4mg versus 5mg), frequency of infusion (single use versus multiple monthly doses), longer lag period and a very modest effect on 'cytokine storm' [16, 19]. Initial studies with alendronate and pamidronate showed gain in terms of improvement in symptom score and suppression of bone turnover but their effect on clinical remission was either not found to be significant or not investigated. Systematic reviews [28, 43] and few recent studies suggested that bisphosphonates may not reduce time to remission in patients with active CN or may even increase the time in cast [16, 19]. The current study shows a remarkable decrease in bone resorption markers and consequent increase in BMC with ZL suggesting effective suppression of ongoing osteoclastogenesis. But

no corelation was observed between an increase in BMC and changes in the inflammatory cytokines or BTMs. However, the increase in BMC over short term could be pertinent in the prevention of deformities and fractures over long-term, due to long retention of the drug in bone matrix. Another observation was that monotherapy with anti-resorptive does not seem to be sufficient for clinical resolution as 'cytokine storm' may continue unabated due to a modest immunomodulatory efficacy of ZL. Denosumab, a monoclonal antibody against RANK-L has been shown to decreased time of fracture healing and improve clinical resolution of active CN in an open-label trial using historical controls [22]. Additionally, the use of recombinant parathyroid hormone does not reduce time to resolution or enhance fracture healing, though foot BMC was not studied [44].

Periodic analysis of bone turnover markers (P1NP and CTX) suggests maximum osteolysis at baseline in all groups, in accordance with clinical activity of active CN as demonstrated previously [23]. Both P1NP and CTX declined in the zoledronate and placebo groups on follow-up without any significant difference between the groups, suggesting an overall decrease in bone turnover because of off-loading [23]. On follow-up, maximum bone mass accrual at ROI was noted with zoledronate despite a similar alteration in cytokines as compared to placebo, suggesting its direct beneficial effect on bone parameters. A similar improvement in foot BMD has been demonstrated earlier with oral bisphosphonate using DEXA [18]. However, whether the initial gain in BMD persists after the quiescence of clinical activity required to be assessed in long term studies.

Despite few interventions like danosumab [22] associated with an enhanced fracture healing and others like zoledronate [19] and recombinant parathyroid hormone [43] not shown to enhance fracture healing over short duration, the long-term outcomes are not studied. Teriparatide has also been shown to increase foot bone BMD in patients of chronic CN of foot [31], but fracture prevention efficacy over long duration is not known. The outcomes of foot bone fracture, incident deformities, ulcers and subsequent amputation are the patient-important outcomes that need attention for amputation prevention. Charcot foot is associated with an increased prevalence of foot deformities noticed in one-third of patients and limb amputation rate of 15.6% when followed for a period of 5 years [30]. However, in the present study, we observed incident deformities of foot only in 8.3% patients, foot ulcer (8.3%), recurrence of acute CN in one patient and no amputations over a follow up of four years irrespective of initial intervention. The lower incidence of foot deformities in the present study could be due to close follow up, reinforcement for appropriate offloading and modified footwear and counselling for foot care practices.

Adverse events were noted in all three groups. Patients in the zoledronate group developed a transient flu-like reaction, which recovered with the use of analgesics and supportive care. The incidence of flu-like reaction in the current study was higher than previously reported (31.6%) where any one component of acute phase response (pyrexia, myalgia, headache, arthralgia, influenza-like symptoms) was considered [33]. Two patients (n = 2, 16%) developed acute kidney injury after the second dose of zoledronate which is comparable to the incidence following standard dosing regimen of bisphosphonates (8 to 15%) [45]. Cast-related tissue injury was observed in two patients from either group with literature evidence showing an overall incidence of 5.7% [46]. A significant proportion of the patients in the methylprednisolone group developed worsening of glycemic profile that was managed by intensification of subcutaneous insulin therapy.

The present study and the results of previous study [23] suggest that suppressing 'cytokine storm' by a potent anti-inflammatory agent may not help in clinical resolution as ongoing cytokine-mediated osteolysis may continue unabated. On the other hand, cytokine-independent activation of RANKL also requires an optimal intervention in addition to anti-resorptives

to effectively suppress osteoclastogenesis. Combined therapy with anti-inflammatory drugs which do not have detrimental effects on bone health (etanercept, infliximab, anakinra) along with agents that can effectively suppress osteoclastogenesis (zoledronate, denosumab) may be an exciting area of research in the future.

The strengths of the current study include the use of an anti-inflammatory agent in active CN, an RCT design, homogenised use of standard-of-care (TCC) in all and precisely defined criteria for remission of active CN. Longer follow-up following intervention provided more patient centric information like fractures, deformities and recurrences, keeping in mind the anticipated benefit of increased BMC due to tendency for long retention of bisphosphonates in the skeleton. The observations from the present study enable proposition of newer aspects of etiopathogenesis of active CN. We perceived certain limitations of the current study that RANKL and anti-inflammatory cytokines (IL-4, IL-10) were not measured, systemic sample than a dorsal venous arch sample and lack of in-vitro assessment of bone biopsy sample would have been useful.

## Conclusion

The current study demonstrates that bisphosphonate use for acute Charcot foot is associated with an increase in foot bone mineral content, but long-term incidence of deformity and amputation are similar amongst interventions. Newer insights into pathophysiology may pave the way for novel therapeutic targets into the still enigmatic Charcot neuroarthropathy.

## Acknowledgments

We would like to thank Miss Priya for assistance in data collection.

## Author Contributions

**Conceptualization:** Ashu Rastogi, Mahesh Prakash.

**Data curation:** Liza Das, Ashu Rastogi, Mahesh Prakash.

**Formal analysis:** Liza Das, Ashu Rastogi, Mahesh Prakash.

**Investigation:** Liza Das, Ashu Rastogi.

**Methodology:** Ashu Rastogi, Edward B. Jude, Mahesh Prakash.

**Project administration:** Ashu Rastogi.

**Resources:** Ashu Rastogi.

**Supervision:** Edward B. Jude, Pinaki Dutta, Anil Bhansali.

**Validation:** Anil Bhansali.

**Writing – original draft:** Liza Das, Ashu Rastogi.

**Writing – review & editing:** Ashu Rastogi, Edward B. Jude, Mahesh Prakash, Pinaki Dutta, Anil Bhansali.

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
