## [Decision Letter · Decision Letter 0]

29 Jun 2021

PONE-D-21-15543

Long-Term Foot Outcomes Following Differential Abatement of Inflammation and Osteoclastogenesis for Active Charcot Neuroarthropathy

PLOS ONE

Dear Dr. Rastogi,

Thank you for submitting your manuscript to PLOS ONE. After careful consideration, we feel that it has merit but does not fully meet PLOS ONE’s publication criteria as it currently stands. Therefore, we invite you to submit a revised version of the manuscript that addresses the points raised during the review process.

I believe there are data which are novel and that need to be further analysed and separated from previously published data in Diabetes Care.

A rebuttal letter that responds to each point raised by the academic editor and reviewer(s). You should upload this letter = as a separate file labeled 'Response to Reviewers'.A marked-up copy of your manuscript that highlights changes made to the original version. You should upload this as a separate file labeled 'Revised Manuscript with Track Changes'.An unmarked version of your revised paper without tracked changes. You should upload this as a separate file labeled 'Manuscript'.

We look forward to receiving your revised manuscript.

Kind regards,

Rayaz A Malik, MBChB, PhD

Academic Editor

PLOS ONE

Additional Editor Comments:

I believe there is merit in publishing some aspects of your study.

However, the major concern raised by reviewer 2 is the overlap with data already published recently by this group in Diabetes Care.

Saying that I think there are limited interventional trial data in studies of the treatment of Charcot and there are data here which could be better presented in a revision addressing the concerns of both reviewers and removing the overlap with the previously published data (Diabetes Care Volume 42, December 2019 e185) focusing on the novel aspects.

1. Additional information on the long-term outcomes of the Charcot foot after a 3 year-follow up (which is new) needs to be emphasized and detail provided on the methodology and data collection.

2. Data on bone mineral content measured at initiation after 6 months of study therapy requires a more detailed analysis and the association with the rate of reduction of BMC and the rate of change of bone turnover markers and interaction with markers of inflammation assessed.

3. Figure 5 illustrates some of the possible drivers of increased osteoclastic activity in the Charcot foot which could be attributed to the direct effect of methylprednisolone on osteoclasts (J Bone Miner Res. 2010 Oct;25(10):2184-92. doi: 10.1002/jbmr.113.).

Journal Requirements:

2. We noted in your submission details that a portion of your manuscript may have been presented or published elsewhere.

[Yes, The data for inflammatory markers and bone turnover markers  at 6 months) were published as interim analysis as Diab Care 2019 Dec;42(12):e185-e186. doi: 10.2337/dc19-1659.

The present mansucripts pertains to bone mineral content and long term outcomes of Charcot foot and teh prior data is presented as figures which are not published elsewhere.]

Reviewers' comments:

Reviewer's Responses to Questions

**Comments to the Author**

1. Is the manuscript technically sound, and do the data support the conclusions?

Reviewer #1: Yes

Reviewer #2: Partly

2. Has the statistical analysis been performed appropriately and rigorously? 

Reviewer #1: Yes

Reviewer #2: Yes

3. Have the authors made all data underlying the findings in their manuscript fully available?

Reviewer #1: Yes

Reviewer #2: Yes

4. Is the manuscript presented in an intelligible fashion and written in standard English?

Reviewer #1: Yes

Reviewer #2: Yes

5. Review Comments to the Author

Reviewer #1: Nice RCT assessing the effect of MT, a biphosphonate and placebo for the treatment of Charcot foot disease. Surrogate and clinical endpoint were assessed at the same time.

The study is well written. However, it does not add much to the literature since findings are already reported in meta-analysis and similar studies. However, it would have been interesting if they added anti-TNF alpha in one of the arms.

Major comments:

1- The follow-up is unclear. How often

If the follow-up is regular, why not to assess the difference using a Kaplan-Meiyer curve which consists of time-to-first event (remission). If any HR is significant, it could be adjusted using baseline characteristics and diabetes parameters using a cox regression model.

2- How was "remission" assessed?

Minor comments:

1- Add number of patients, age and sex in the abstract.

Reviewer #2: Dear authors,

The aim of the study is to explore the effect of anti-inflammatory or anti-resorptive agents for remission of active CN foot on bone mineral on content (BMC) of the active Charcot foot as well as the effect of these therapies on the consequent long-term outcomes of foot deformities, fractures and amputation in people with Charcot foot.

The results of the efficacy of methylprednisolone and zoledronate versus placebo in the management of the Charcot foot have been already published by the same group and it is disappointing to see that this manuscript contains a lot of already published data, although presented in a slightly different context. The paper offers some good points for discussion and figure 5 illustrates some of the possible drivers of increased osteoclastic activity in the Charcot foot. One aspect that could be added is the direct effect of methylprednisolone on osteoclasts (J Bone Miner Res. 2010 Oct;25(10):2184-92. doi: 10.1002/jbmr.113.), which has not been considered when designing this therapy. Glucocorticoids maintain osteoclasts in a prolonged continuous resorption mode, therefore the reported longer duration of casting in patients treated with this agent is not surprising, (despite the effect of steroids to inhibit TNF-alpha primed osteoclastogenesis).

The submitted manuscript to a great extent overlaps with data from a recent intervention trial, which is already published by the same group (Diabetes Care Volume 42, December 2019 e185). The additional information on the long-term outcomes of the Charcot foot after a 3 year-follow up (which is new) does not stand out and is almost left to the end of the result section lacking a lot of detail in methodology and data collection.

A further new aspect of the current submission is the data on bone mineral content measured at initiation after 6 months of study therapy. This could be potentially of interest. However, it requires a more detailed analysis and perhaps association should have been sought between the rate of reduction of BMC and the rate of change of bone turnover markers and the interaction with markers of inflammation.

These new observations might be of interest to be submitted as a commentary or short report. However, the way the current manuscript is presented, these new additions are almost absorbed by the data of the actual (already published) clinical trial.

Thus, in my opinion, this manuscript does not contain sufficient new data for a further research paper.

6. PLOS authors have the option to publish the peer review history of their article (what does this mean?). If published, this will include your full peer review and any attached files.

Reviewer #1: No

Reviewer #2: No

---

## [Author Response · Author response to Decision Letter 0]

1 Oct 2021

Sir,

 We sincerely thank you and the reviewers for their time in reviewing the manuscript and providing valuable comments and suggestions. We have modified the manuscript accordingly and the changes in the revised manuscript are highlighted BLUE. 

Also, please find below the point-by-point reply to the reviewers’ comments.

Reviewer #1: Nice RCT assessing the effect of MT, a biphosphonate and placebo for the treatment of Charcot foot disease. Surrogate and clinical endpoint were assessed at the same time.

The study is well written. However, it does not add much to the literature since findings are already reported in meta-analysis and similar studies. However, it would have been interesting if they added anti-TNF alpha in one of the arms.

Reply: We sincerely thank you for your appreciation and the suggestions. We agree that TNF alpha (Etanercept) could have been added in one of the arms. Etanercept use was not planned (not available in our country at the time of study initiation). However, we will consider future studies with etanercept. 

Major comments:

1- The follow-up is unclear. How often

If the follow-up is regular, why not to assess the difference using a Kaplan-Meier curve which consists of time-to-first event (remission). If any HR is significant, it could be adjusted using baseline characteristics and diabetes parameters using a cox regression model.

Reply: The follow up details following remission of active Charcot is provided in the revised manuscript (patients were recalled fortnightly). We performed Kaplan-Meier analysis for the time to remission and subsequently analysed predictors of remission by Cox regression model. The details are presented in the revised manuscript (Results section).

2- How was "remission" assessed?

Reply: Remission criteria for active charcot are mentioned in the method section of revised manuscript.

Minor comments:

1- Add number of patients, age and sex in the abstract.

Reply: Thanks for the suggestion. The number of patients, age and sex are added in the revised abstract.

Reviewer #2: Dear authors,

The aim of the study is to explore the effect of anti-inflammatory or anti-resorptive agents for remission of active CN foot on bone mineral on content (BMC) of the active Charcot foot as well as the effect of these therapies on the consequent long-term outcomes of foot deformities, fractures and amputation in people with Charcot foot.

The results of the efficacy of methylprednisolone and zoledronate versus placebo in the management of the Charcot foot have been already published by the same group and it is disappointing to see that this manuscript contains a lot of already published data, although presented in a slightly different context. The paper offers some good points for discussion and figure 5 illustrates some of the possible drivers of increased osteoclastic activity in the Charcot foot. One aspect that could be added is the direct effect of methylprednisolone on osteoclasts (J Bone Miner Res. 2010 Oct;25(10):2184-92. doi: 10.1002/jbmr.113.), which has not been considered when designing this therapy. Glucocorticoids maintain osteoclasts in a prolonged continuous resorption mode, therefore the reported longer duration of casting in patients treated with this agent is not surprising, (despite the effect of steroids to inhibit TNF-alpha primed osteoclastogenesis).

Reply: We thank you for the suggestion. We agree regarding the direct effect of steroids on osteoclasts. However, premise of the study was that the inciting event of inflammation and inflammatory cytokine mediated activation of RANKL could be curbed with steroids so as to have earlier remission. The effect of BMC with steroids was an expected outcome on short term that is the decrease in BMD of foot bone. The same is discussed in the revised manuscript.

The submitted manuscript to a great extent overlaps with data from a recent intervention trial, which is already published by the same group (Diabetes Care Volume 42, December 2019 e185). The additional information on the long-term outcomes of the Charcot foot after a 3 year-follow up (which is new) does not stand out and is almost left to the end of the result section lacking a lot of detail in methodology and data collection.

Reply: We accept that the present study describes the acute charcot remission with the three therapies, that is curtailed in the revised manuscript as suggested in view of prior publication. However, the revised manuscript also entails the long-term outcomes after initial intervention and the Kaplan Meier analysis and predictors of remission. The method, result and discussion section is modified accordingly.

A further new aspect of the current submission is the data on bone mineral content measured at initiation after 6 months of study therapy. This could be potentially of interest. However, it requires a more detailed analysis and perhaps association should have been sought between the rate of reduction of BMC and the rate of change of bone turnover markers and the interaction with markers of inflammation.

These new observations might be of interest to be submitted as a commentary or short report. However, the way the current manuscript is presented, these new additions are almost absorbed by the data of the actual (already published) clinical trial.

Reply: We thank you for the suggestions. We have done a corelation analysis between change in BMC and the changes in the inflammatory markers and BTMs. The same are presented in the result section of revised manuscript.

We hope that the revised manuscript may be considered for publication.

Sincerely,

Ashu Rastogi

Deptt. of Endocrinology

PGIMER, Chandigarh

INDIA

---

## [Editor Report · Decision Letter 1]

18 Oct 2021

Long-Term Foot Outcomes Following Differential Abatement of Inflammation and Osteoclastogenesis for Active Charcot Neuroarthropathy in Diabetes Mellitus

PONE-D-21-15543R1

Dear Dr. Jude,

We’re pleased to inform you that your manuscript has been judged scientifically suitable for publication and will be formally accepted for publication once it meets all outstanding technical requirements.

Kind regards,

Rayaz A Malik, MBChB, PhD

Academic Editor

PLOS ONE

Additional Editor Comments (optional):

Thank you for addressing the concerns of the reviewers and revising the manuscript appropriately.
---

## [Editor Report · Acceptance letter]

27 Oct 2021

PONE-D-21-15543R1 

Long-Term Foot Outcomes Following Differential Abatement of Inflammation and Osteoclastogenesis for Active Charcot Neuroarthropathy in Diabetes Mellitus 

Dear Dr. Jude:

I'm pleased to inform you that your manuscript has been deemed suitable for publication in PLOS ONE. Congratulations! Your manuscript is now with our production department. 

Kind regards, 

on behalf of

Professor Rayaz A Malik 

Academic Editor

PLOS ONE